# Modified Ant Colony Optimization as a Means for Evaluating the Variants of the City Railway Underground Section

**DOI:** 10.3390/ijerph20064960

**Published:** 2023-03-11

**Authors:** Mariusz Korzeń, Maciej Kruszyna

**Affiliations:** Faculty of Civil Engineering, Wrocław University of Science and Technology (Politechnika Wrocławska), 50-370 Wrocław, Poland

**Keywords:** agglomeration railway, modified ant colony optimization, evaluation of railway tunnel variants, choosing of options

## Abstract

The railway is one of the most energy-efficient modes of transport, helping to enhance the environment and public health in cities and agglomerations. In this paper, the authors raise the issue of the construction of an underground railway route in Wrocław (Poland) to allow the organization of the suburban rail system in the agglomeration. There are many concepts for the construction of this route, but so far none has been realized. Therefore, it is important to design the route properly. Here, five options for this tunnel are considered and evaluated. To make such an evaluation, the authors construct a modified ant colony optimization algorithm (ACO). The “classic” algorithm considers the determination of the shortest route. The modification of the algorithm will allow a more accurate analysis of the issue, taking into account more parameters than just the length of the route. These are the location of traffic generators in the city center, the number of inhabitants neighboring the stations, and the number of tram or bus lines integrated with the railway. The presented method and exemplary case study should allow for the evaluation, introduction, or development of the city railway.

## 1. Introduction

The issue of network planning and the transportation planning process are very complex processes aimed at the functional design of transportation supply system projects [1]. It is important on a global scale, but also on a regional scale [2]. One of the significant elements of transportation planning is public transportation, which is an integral part of any agglomeration. It must provide a reliable, convenient, and economical service environment with high capacity. Its efficiency is important for many aspects of the functioning of the entire urban area, and for public health. Unfortunately, many problems can still be encountered that hinder the efficient transportation of people and goods. The problem of improving traffic conditions in cities is the subject of many studies, which differ in cases and research methods. In [3,4,5,6,7], the authors attempt to analyze and compare the planning of urban transportation systems, for example through a preliminary assessment of current technical and economic solutions resulting from the introduction of systems of the latest design. In order to evaluate the development of sustainable public transportation in [8], the authors proposed a multi-criteria coordinated model based on economic, social and environmental data. The study [9] develops a quick evaluation model of public transport efficiency to analyze the impact of changes in road design and policies that may affect public transportation services. Investment challenges pertaining to the achievement of the goals of the Mobility Policy were studied in [10]. The problem of accessibility to cities is analyzed in [11] and in [12]. Here, short-term passenger flow prediction is important [13].

Heuristic methods are often used to solve problems of a transportation nature. In [14], the authors propose a heuristic method for designing a transit route network, taking into account a number of important parameters such as budget constraints, level-of-service standards, and the attractiveness of transit routes. Additionally, in [15,16], a genetic algorithm is used to optimize public transportation routes. There are also attempts to optimize entire transit networks [17,18,19]. In [20], the authors use a genetic algorithm to solve a network design problem in an urban area in a multi-criteria manner. Both network layout and connection capacity are optimized. Particle Swarm Optimization (PSO) is also used for optimization, and an example of its application is described in [21,22]. It is also possible to combine heuristic methods. One then speaks of hyper-heuristics, which are also used to solve transportation problems [23]. An interesting heuristic tool for modeling the use of suburban railways is presented in [24]. There are many possible heuristic methods that work to compare different methods with each other [25]. Consideration of the big data sets is important in planning and modeling transportation systems or other aspects of urban understanding [26]. Therefore, artificial intelligence is used for the better functioning of the system.

Particular attention should be paid to transportation networks whose main means of public transport are railroads. Railroads are, or should be, one of several integrated elements that work together at different levels [27,28]. The aspect of integration and cooperation of different transport means is discussed in many studies [29,30]. It is important to keep in mind a properly tailored timetable that is adapted to changing demand [31]. One way to achieve integration between transportation modes could be the concept of sustainable Mobility as a Service (MaaS). This is a new approach in transportation systems that gives users the opportunity to use different transportation services as a single option, through the use of digital platforms and integrated design [32,33,34]. Some researchers highlight the problem of a bad influence of railway lines on the surrounding area and for residents [35,36]. A literature review of applications of artificial intelligence to rail systems is described in [37]. In medium-sized cities (up to one million residents in a metropolitan area), the basis of public transportation is a tramway or tramway in combination with a suburban railroad. In this case, the suburban railroad transports travelers from peripheral areas to the central part of the city. The journey is then continued by tramway. By planning or introducing an urban railway, two main aspects should be considered. The first one is the decision about the location of stops and the distances between them [38]. The second one is the connection of stops with surroundings, the classification of stops, and the creation of an additional program to shape a mobility node in an aspect of Transit Oriented Development (TOD) [39,40,41].

The specific heuristic method, namely ant colony optimization (ACO) [42], is used in this paper. This method is used to solve many optimization problems. For example, it is used to determine the optimal route [43,44]. It is also possible to use ACO to optimize the entire transportation network [45,46,47] and to plan multimodal routes [48]. Typically, the ACO without modification allows for determining the shortest route. Therefore, in this paper, the algorithm is modified. In [49], the authors use an improved ACO to solve the vehicle routing problem (VRP). The improved model has a better simulation effect and can use traffic resource allocation, enabling the transportation system to achieve a comprehensive optimization of time, cost, and the number of accidents. In [50] an improved ant colony optimization (IACO) is proposed, which has a new strategy for updating increased pheromones, called the ant weight strategy, and a mutation for solving VRP. The modification of the algorithm based on pheromone updating is described by the authors in [51], so that the enhanced ant colony algorithm is given to solve VRP in a time-dependent network. ACO is also used to solve the traveling salesman problem (TSP) [52], and its modification, for example, by using a mutation operator [53], allows for obtaining more efficient solutions.

The conducted literature studies show that currently known and used optimization methods will not allow the obtaining of authoritative results in the considered issue. This is due to the specifics of the issue, which are different from those in the cited references, as well as restrictions in the obtaining of data. Therefore, it is necessary to adapt the known methods to the analyzed problem and to the available data. In this paper, the modification of the algorithm will allow a more thorough analysis of the issue, taking into account more parameters than just route length. The modification of the algorithm involves taking into account factors related to the number of potential passengers, and increasing accessibility to the network and the attractiveness of areas. The purpose of the study is to see which of the proposed underground rail routes is best, using the modified ACO. The construction of new routes will create a suburban rail network. As a result, it will be possible to improve traffic conditions throughout the metropolitan area.

Not every city operates a suburban rail network. One such city is Wrocław (Poland). The basic form of public transport in this city is a tramway. The available infrastructure allows for the development of a suburban rail network in Wrocław and its surroundings. The problem is the limited capacity of the current cross-city section. Therefore, it is necessary to create a new route. In the city center, it is best to make it underground. The construction of a tunnel is associated with high costs, so it makes sense to design the route well. Therefore, the case study of this paper (Section 3) is dedicated to Wrocław. Five options for railway tunnels are presented and evaluated using the authors’ method, based on the modified ACO algorithm.

## 2. Methodology

### 2.1. Ant Colony Optimization

Ant colony optimization is a meta-heuristic technique which uses artificial ants to find solutions to combinatorial optimization problems. In nature, a single ant is unable to communicate or effectively hunt for food, but as a group ants have the ability to solve complex problems and successfully find and collect food for their colony. Ants, as they can see very poorly (at a few cm), communicate with each other using a chemical called a pheromone. During migration, the ant secretes a constant amount of pheromone that other ants can follow. Each ant moves at random, but when it encounters a pheromone it must decide whether or not to follow it. If it follows the pheromone, the ant’s own pheromone reinforces the existing path, and the increase in pheromone increases the likelihood that the next ant will take the same path. Therefore, the more ants move along the path, the more attractive the path becomes to the next ants. This affects the probability of choosing the path for the next ant leaving the nest. The more ants are able to travel a shorter route, the faster the pheromone accumulates on shorter routes and longer routes are less enhanced. In this way, the shortest path is determined. This process is showed in Figure 1.

The principle of the algorithm can be described by the following formulas:
Probability of *k*-ants crossing section *ij*:(1)$$p_{ij}^{k}=\frac{\left(\tau_{ij}^{\alpha }\right)\left(\eta_{ij}^{\beta }\right)}{\sum_{z\epsilon allowed_{i}}\left(\tau_{ij}^{\alpha }\right)\left(\eta_{ij}^{\beta }\right)}$$
where$\;\tau_{ij}$ is the amount of pheromone at section *ij*; and $\eta_{ij}$ is the attractiveness of section *ij*. In transport issues:(2)$$\eta_{ij}=\frac{1}{L}$$
where L is the total route to be taken; α is the parameter increasing $\tau_{ij}$. The larger α the more we “trust” the information left by other ants. It is assumed that α ≥ 0. The user of the algorithm can freely control the parameter; β is the parameter increasing $\eta_{ij}$. The larger β is, the more we “trust” our own experience. It is assumed that β ≥ 1. The user of the algorithm can freely control the parameter.Amount of pheromone at section *ij*:(3)$$\tau_{ij}=\left(1-\rho \right)\tau_{ij}+\underset{k}{\sum }\Delta \tau_{ij}^{k}$$
where $\Delta \tau_{ij}^{k}$ is the amount of pheromone left by *k*-ant in the next iteration; ρ is the evaporation coefficient of the pheromone. It takes a value from 0 to 1. When ρ=1 then the pheromone evaporates completely after each iteration.Amount of pheromone left by *k*-ants in the next iteration:(4)$$\Delta \tau_{ij}^{k}=\left\{\begin{array}{c} Q/L_{k}\\ 0 \end{array}\right.$$
where Q is a fixed value indicating the amount of pheromone; $L_{k}$ is the length of the route travelled by *k*-ant.

If the *k*-ant passes through segment *ij*, the value of the coefficient $\Delta \tau_{ij}^{k}$ takes the value $Q/L_{k}$. In other cases, where the ant has not walked the path and has left no pheromone, the coefficient $\Delta \tau_{ij}^{k}$ takes the value 0.

A variation of this algorithm is where the artificial ants are supposed to find the shortest path, but have to cover all the points along the way. In the beginning, the ants’ behavior is dispersed and they travel along different paths, but they cover all points. In further iterations, the ants are influenced by the pheromone left by other ants. Finally, the ants will follow only one route (the optimal one). This is a similar task to the traveling salesman problem. The difference is that, in the case under consideration, we assume that the ants do not return to the starting point, they just have to go through all the points. This process is shown in Figure 2.

### 2.2. Modification of the Algorithm

For the purpose of improving the performance of the algorithm, a modification has been applied. The length between points will be changed. Parameters will be added to make the paths taken easier or harder. As a result, the algorithm calculates the effort an ant has to put in to travel the path and, consequently, the entire route. The modified ACO is implemented in the python programming environment. The program works on the basis of an effort matrix (5):(5)$$E=\left[\begin{array}{ccc} E_{11} & \cdots & E_{1j}\\ \vdots & \ddots & \vdots \\ E_{i1} & \cdots & E_{ij} \end{array}\right]$$

This matrix is square and symmetric. All modifications will be made to the *E_ij_* coefficients of the matrix *E*. Each coefficient of the matrix determines the effort needed to travel to each point (station), and is defined by the formula (6):(6)$$E_{ij}=\frac{L_{ij}}{L_{c}}\cdot \left(1-\frac{I_{ij}}{I_{c}}\right)\cdot \frac{1}{A_{ij}}\cdot \frac{S_{max}}{S_{c}}$$
where $E_{ij}$ is the ant’s effort between stations *ij*; $L_{ij}$ is the distance between stations *ij*; $L_{c}$ is the total length of tunnel; $I_{ij}$ is the approximate number of inhabitants located in the area of station *ij*; $I_{c}$ is the total number of inhabitants located in the tunnel area; $A_{ij}$ is the attractiveness index; $S_{max}$ is the maximum number of new areas with access to the rail network among all proposed routes; $S_{c}$ is the total number of new areas with access to the railroad network.

As the effort increases, the artificial ants have to put in more energy to travel the route. Consequently, the greater the value of E, the worse the route involved. The distance between stations and the total length of the route is known on the basis of the geometry. The estimated number of inhabitants living in the area near the station is determined on the basis of GIS data for the city of Wrocław [54]. A map of the city shows the population density of the individual city regions. In this paper, the station area is assumed to be a circle with a radius of 300 m. On this basis, $I_{ij}$ has a value (7):(7)$$I_{ij}=\frac{I_{i}+I_{j}}{2}$$
The values $I_{i}$ and $I_{j}$ denote the number of inhabitants living in the area of the planned station. Taking the entire population within a station is a simplified variable representing potential transportation demand. The adoption of this simplification is due to the fact that public transportation is significantly emphasized in mobility plans. As a result, it is assumed that a large share of trips should be made by the most efficient means of transportation, which is rail. The attractiveness index is defined by the formula (8):(8)$$A_{ij}=\frac{A_{i}+A_{j}}{2}$$
The $A_{i}$ and $A_{j}$ values are the number of bus and tram lines passing through a given junction within the planned station. The values are read on the basis of timetables available on the website of the operator (i.e., MPK Wrocław). The $S_{c}$ value determines the total number of new areas that will gain access to the rail network. Areas that already have rail access are not included in the $S_{c}$ value. In this case, the number of proposed stations that are in areas without access to the rail network should be given. The $S_{max}/S_{c}$ value allows you to compare the routes proposed in this paper in terms of the increase in rail accessibility in the city center. The more new stations, the more rail accessibility in the center increases, and thus the demand for rail travel increases.

The next step is to recalculate all the $E_{ij}$ coefficients of the E matrix for the proposed routes and enter the data into Python. In addition, the algorithm assumes that the evaporation coefficient of the pheromone factor ρ and the factors α and β (in this case ρ=0.5, α=1 and β=2).

## 3. Case study

### 3.1. Research Area

Wrocław is a city located in south-western Poland in Lower Silesia. The city has about 650,000 inhabitants. The entire agglomeration has approximately one million inhabitants. Due to its long history, the Wrocław Railway Junction (WRJ) is one of the most extensive track systems in Poland. Ten railway lines converge in Wrocław, covering most of the city’s area. The current state of the WRJ in graphic form is shown in Figure 3. The center is marked by a red circle. Analyzing the WRJ, it can be concluded that the railway lines encircle the city center but do not cross it. Consequently, the railway is not an attractive means of transport for the inhabitants of the area. To change this, a railway tunnel running under the city center should be designed.

The question arises as to how to route the tunnel. Firstly, the starting point of the tunnel should be considered. The best place for this is the disused Świebodzki Railway Station. The area around the station is unused and this will allow the tunnel to start without problems. Next, the route of the tunnel through the city center should be considered. The route is best dependent on the traffic generators that are located in the center. Such places could include railway stations, the market square, major interchanges, academic centers, business centers and historic sites. In the case of Wrocław, the location of the new diameter should be dictated by the traffic generators located within the city, and these are:Main Railway Station—interchange between different modes of transport;Old Town/The Market Square—area generating significant tourist traffic;Dominikański Square—large interchange by the Old Town;Bema Square—interchange towards the northern part of the city;Powstańców Warszawy Square—to be developed in the future as an important business and service location;Grunwaldzki Square—interchange in the city. In the vicinity there are many academic centers;Centennial Hall/Zoological garden—places that generate a significant amount of tourist traffic;Nadodrze Railway Station—interchange between different modes of transport;Crossroads at Wrocław Arcades—interchange towards the southern part of the city.

The locations of the identified sites are shown in Figure 4. These are densely built-up areas, so it is only possible to run the railway line in a tunnel. Ideally, all of the sites listed should have direct access to the line. This will significantly increase the attractiveness of rail transport in the city and agglomeration.

### 3.2. Description of Survey

The authors propose five routes based on the traffic generators discussed in the previous section (Figure 4). Route one (Figure 5) has an east–west direction. The route ends on the eastern side at Swojczyce Railway Station. Route two (Figure 6) is very similar to route one. The difference is in part of the route, as the Dominikański Square and the Powstańców Warszawy Square are skipped. Instead, there is a station at the Bema Square, so that the northern part of the city has access to the rail network. Route three (Figure 7) is a link between Świebodzki Station and Main Railway Station. Connecting the stations will enable convenient transfers between stations and between regional, suburban and long-distance trains. The route is short (2.5 km) and there is only one intermediate station in the area of the junction by the Wrocław Arcades. Unfortunately, the route does not cover most of the city center. The fourth route (Figure 8) is similar to route three. It connects Świebodzki Railway Station and the Main Railway Station. The route is not in the direct area of the Main Railway Station. From the south, the route connects to the station’s track system. This will make it easier to make a tunnel. The route passes through the Old Town and the Market Square. Thus, despite the short length of the route, an increase in accessibility to the rail network in the center is achieved. The fifth route (Figure 9) has a north–south alignment. It starts from the north in the area of Nadodrze Railway Station, but not directly at the station. On the other hand, the route ends at the Main Railway Station. At present, the stations are connected by tracks but they encircle the city center. Figure 5, Figure 6, Figure 7 and Figure 8 show the route and station locations. Stations located in the area of important traffic generators are described in red. Other stations are described in black. A detailed description of the routes is provided in Table 1.

In the next step, the coefficients $E_{ij}$ are determined from Formula (6) and the effort matrices E are created. The obtained matrices are inputted into Python. The program calculates the effort that the artificial ants put into traveling each route. The resulting values will be the basis for comparing the proposed routes. The Python code for the considered ACO algorithm is included in the Appendix A. The code for route number one is shown. For the other routes, only the effort matrix E is different in the code.

## 4. Results

Table 2, Table 3, Table 4, Table 5 and Table 6 show the effort matrices determined for the proposed routes. The numbers of columns and rows in the matrices correspond to the numbers of stations located on each route (Figure 5, Figure 6, Figure 7 and Figure 8). The coefficients of the matrix $E_{ij}$ on the diagonal are zero, since according to Formula (6) the distance $L_{ij}$ between the same stations is zero.

Table 2 presents the $E_{1}$ effort matrix for route one. It is the largest of the matrices because the route it describes has the most stations. The matrix reaches its largest value for the pairs of stations 2 and 9, and reaches a value of 23.5. This means that artificial ants to cover the route from station 2 to station 9 (without taking into account stations 3–8) have to incur an effort of 23.5. The elements of this value include: the distance between stations ($L_{29}\;$= 7702 m), the total length of route number one ($L_{c}\;$= 8383 m), the approximate number of inhabitants between stations 2 and 9 ($I_{29\;}$= 4,500 people (calculated as the average of the two areas)), the total number of inhabitants located in the area of route number one ($I_{c}\;$= 43,000 people), the attractiveness index ($A_{29}\;$= 3.5 (calculated as the average of the number of bus and tramway lines running through station areas 2 and 9)), the number of new areas with rail access ($S_{c}$ = 9) and the maximum number of new areas with rail access ($S_{max}\;$= 9). The number one route has the most new areas proposed, so $S_{max}$ = $S_{c}$. The $S_{max}$/$S_{c}$ factor is constant for a given route. In order to obtain clearer results, all $E_{ij}$ coefficients are multiplied 100 times. The highest value for the pair of stations 2 and 9 is mainly due to the distance between stations and the small number of buses and the tramway running in the area of station number 9. The lowest value of the $E_{ij}$ coefficient is obtained for the pair of stations 2 and 3, and is 0.7. This is due to the short distance between stations, the large number of inhabitants in the area of the stations, and the large number of buses and the tramway running in the area of stations 2 and 3.

Matrix $E_{2}$ (Table 3) has similar characteristics to matrix $E_{1}$. The routes for which matrices $E_{1}$ and $E_{2}$ are determined have similar lengths and numbers of stations. They differ slightly in the routing of the downtown area. The values in matrix $E_{2}$ oscillate between 0.9 and 26.9, and are similar to those obtained in matrix $E_{1}$, although the maximum value for the $E_{2}$ matrix is larger than for the $E_{1}$ matrix. It is obtained for the pair of stations 2 and 8. The reasons for such a high value are analogous to those for matrix $E_{1}$. In addition, for matrix $E_{2}$, the $S_{max}$/$S_{c}$ quotient is larger than for matrix $E_{1}$, which further increases the effort required for the route.

Matrices $E_{3}$ (Table 4) and $E_{4}$ (Table 5) have similar characteristics. The routes for which matrices $E_{3}$ and $E_{4}$ are determined have similar routes, numbers of stations and length. The value of $S_{max}$/$S_{c}$ for route number three is the largest and increases the effort needed to cover the route by four times (compared to route number 1). This can be seen in the values of the $E_{ij}$ coefficients. The lowest value in matrix $E_{3}$ is 6.8, and the highest is 15.7. For matrix $E_{4}$, the coefficients oscillate between 5.2 and 20.1. It is worth noting that, for matrix $E_{4}$, the value of $S_{max}$/$S_{c}$ increases the $E_{ij}$ coefficients twice, which differentiates between routes three and four. The higher coefficients of the $E_{4}$ matrix are mainly due to the small number of buses and tramways running in the area of station number 4.

Table 6 presents the $E_{5}$ effort matrix for route five. The coefficients of the $E_{5}$ matrix oscillate between 4.6 and 18.4. The lowest indicated value is obtained for the pair of stations 2 and 3 and the pair of stations 3 and 4. This is due to the high attractiveness index, which reduces the effort needed to travel the route. On the other hand, the high $S_{max}$/$S_{c}$ value of 2.67 increases the effort needed to travel the route. The $E_{5}$ matrix has the same size as the $E_{4}$ matrix. Due to the totally different characteristics of the routes for which matrices $E_{5}$ and $E_{4}$ are determined, the values in these matrices are different.

Table 7 summarizes the proposed routes along with the final results. The values obtained are dimensionless to determine the effort each ant had to put in to cover each route. The least effort, and thus the best, is the third route. This is the route connecting the Świebodzki Railway Station and the Main Railway Station. To cover the second-best route (route number one), the ants had to put in 15% more energy compared to the best route. A similar result is achieved by route four. It takes about 17% more effort to cover it. To cover the fourth-best route (route number two), the ants need almost 30% more energy. The weakest route is route number four. To cover it, the ants need more than 40% more energy compared to the best route.

## 5. Discussion

The paper applies a modified ACO to evaluate rail underground route proposals. The task of the algorithm was to verify the routes to determine which one would work best as a new cross-country route for a future suburban railroad. Five routes were evaluated. Routes number 1 and 2 are the longest and run east–west. They provide the greatest accessibility to the rail network in the center but are the most expensive to build. Route number 3 connects Świebodzki Railway Station and the Main Railway Station. Its construction will relieve the current cross-city section. The route is the shortest of those proposed in the paper. Therefore, it will not significantly increase rail accessibility in the center. Route four is similar to route three, but with the location of the station closer to the city center it allows the relief of congestion on the existing cross-city section. In addition, the route allows increased accessibility to the rail network in the center. Route number 5 is routed in a north–south direction and connects Nadodrze Railway Station with the Main Railway Station. Currently, there is already such a connection on the WRJ, but it circulates around the city center. The new route runs through the center. The criteria analyzed in the evaluation are the length of the route, the increase in accessibility to the rail network in the city center, the number of potential new passengers and the attractiveness of the areas.

ACO is implemented in the Python programming language. For each route, the algorithm calculated the effort the artificial ants needed to put in to travel the route. As a result, the ants put in the least effort to travel route number 3. Therefore, based on the results obtained, it can be concluded that route number 3 is the best to implement.

Route number 3 is a good supplement to the WRJ and will improve rail traffic in the city center. The route will make it possible in the future to create an efficient suburban rail network. In the opinion of the authors, the route, which according to the algorithm is the best, will not meet an important objective and will not significantly increase accessibility to the rail network in the city center. This may result in a small increase in the attractiveness of the railroad, and thus a small number of new travelers.

## 6. Conclusions

The applied methodology of the research allows the evaluation of the proposal of underground railroad routes in Wrocław. As a result, we found the effort that artificial ants had to put in to traverse each one. For route one, the effort was 30.2; for route two it was 33.9; for route three it was 26.2 (the lowest value); for route four it was 30.6; and for route five it was 37.3 (the highest value). The results obtained show the differences in the effort of the ants, and thus the differences between the different route proposals. Based on the results, we have identified a route that could be a main component of the suburban rail network in the future.

The results obtained from the study indicate that there are several issues that require further consideration in future research of the problem under consideration. The authors evaluated a restricted list of routes. Five proposals were considered. This restricts the field of research on the optimal route to only the authors’ proposals. The evaluation process is based on a limited number of criteria. Four different criteria are used in the paper. For a better evaluation, more criteria can be analyzed. The use of criteria based on discrete data is also a problem. The paper also adopts some data simplifications. For example, it was assumed that all inhabitants within the station would only travel by rail. This is quite a simplification, although acceptable for the issue under consideration. In future studies, the model can be expanded to include all modes of transportation (car, transit, bicycle, etc.).

It is also important to consider whether the evaluation criteria considered and their number are sufficient to select the optimal route. The use of ACO (despite the modification of the algorithm) is also based on discrete data. Future research may consider combining the proposed routes into variants to be evaluated. One can also consider increasing the number of evaluation criteria to increase the accuracy of the results. Finally, the research can be based on the use of methods based on continuous data, and other optimization methods from the group of heuristics, such as genetic algorithms, can be tested.

## Figures and Tables

**Figure 1 ijerph-20-04960-f001:**
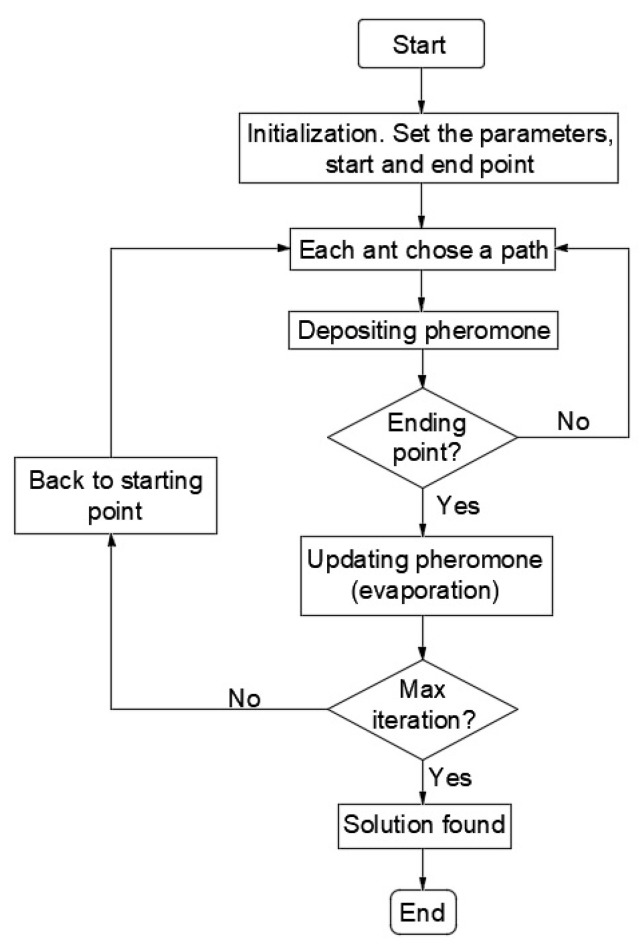
Flowchart of the ACO operation.

**Figure 2 ijerph-20-04960-f002:**
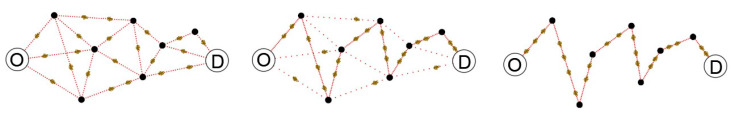
ACO operation scheme in the case study. Indications: O—origin, D—destination.

**Figure 3 ijerph-20-04960-f003:**
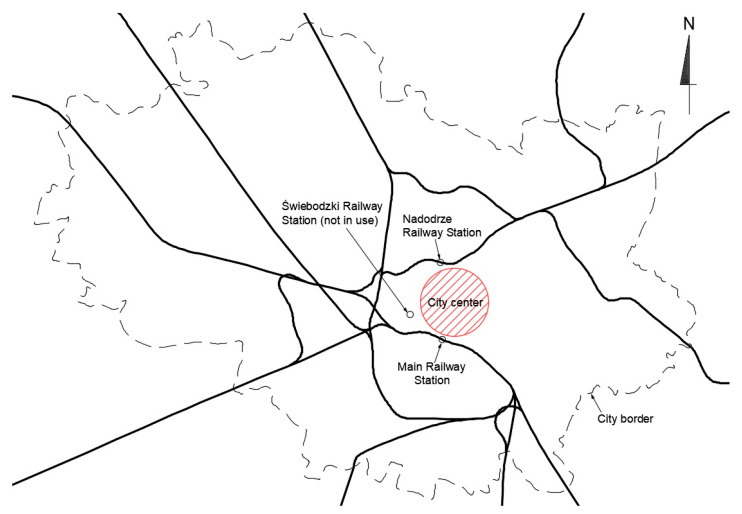
The Wrocław Railway Junction (WRJ).

**Figure 4 ijerph-20-04960-f004:**
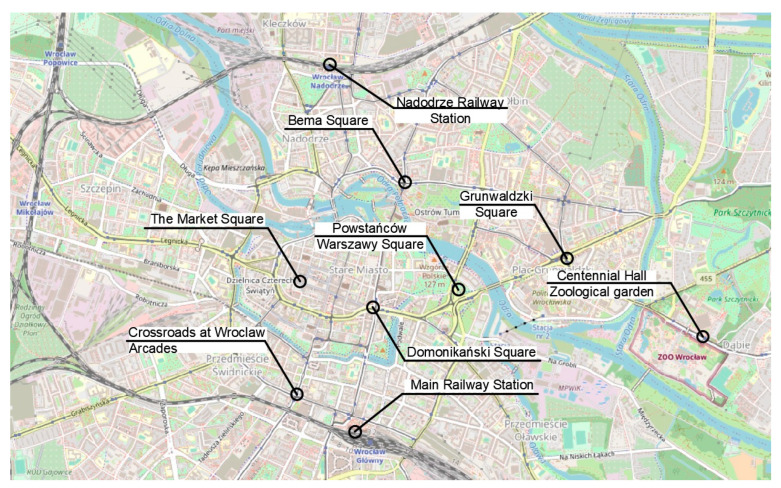
Traffic generators. (Background: openstreetmap.org).

**Figure 5 ijerph-20-04960-f005:**
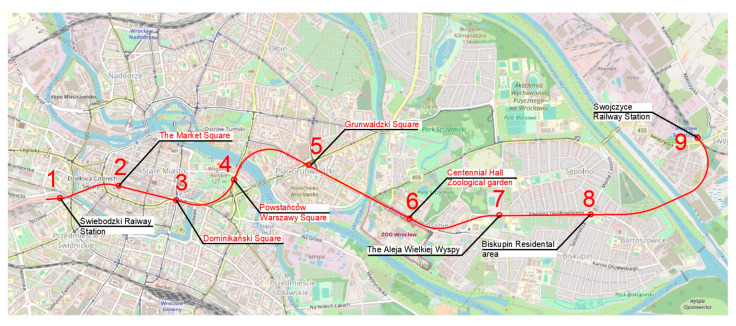
Route one (background: openstreetmap.org).

**Figure 6 ijerph-20-04960-f006:**
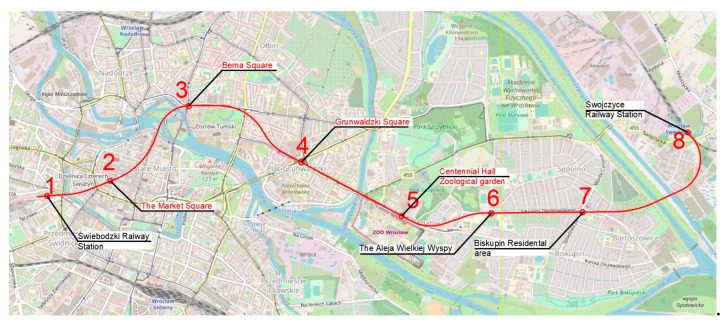
Route two (background: openstreetmap.org).

**Figure 7 ijerph-20-04960-f007:**
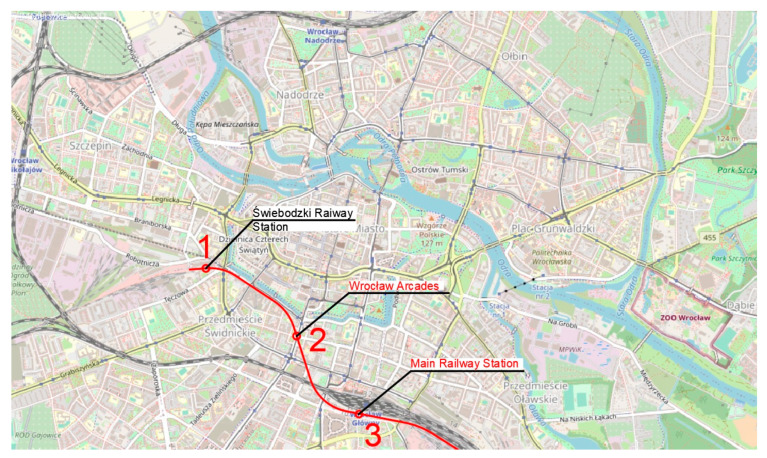
Route three (background: openstreetmap.org).

**Figure 8 ijerph-20-04960-f008:**
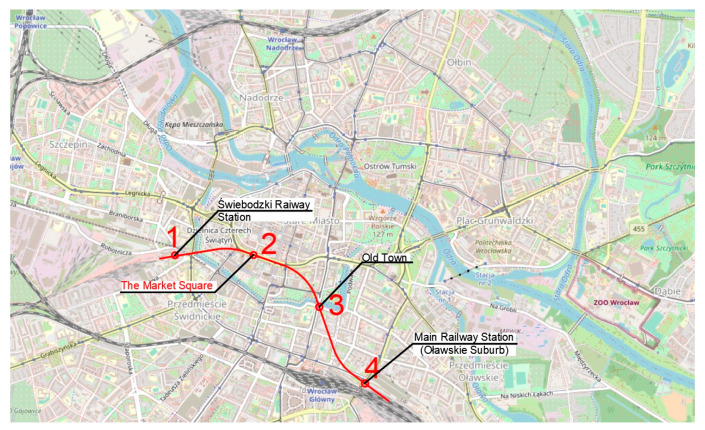
Route four (background: openstreetmap.org).

**Figure 9 ijerph-20-04960-f009:**
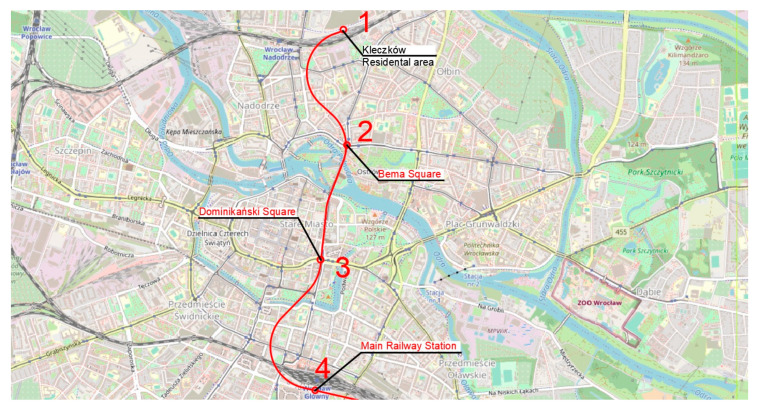
Route five (background: openstreetmap.org).

**Table 1 ijerph-20-04960-t001:** Description of routes.

Route Number	Route Length $\left(L_{c}\right)$	Number of Stations	Number of New Areas with Access to the Railroad Network $\left(S_{c}\right)$	Number of Inhabitants Located in the Tunnel Area $\left(I_{c}\right)$
1	8,382 m	9	8	43,000
2	9,262 m	8	7	44,000
3	1,927 m	3	2	17,000
4	2,447 m	4	4	19,000
5	3,886 m	4	3	20,000

**Table 2 ijerph-20-04960-t002:** Effort matrix $E_{1}$ for route one.

	1	2	3	4	5	6	7	8	9
**1**	0.0	0.9	1.1	2.0	1.9	4.5	7.3	8.2	10.9
**2**	0.9	0.0	0.7	1.9	2.0	6.0	13.4	15.5	23.5
**3**	1.1	0.7	0.0	0.6	0.9	2.5	4.8	5.7	9.1
**4**	2.0	1.9	0.6	0.0	0.7	2.6	4.9	6.2	10.2
**5**	1.9	2.0	0.9	0.7	0.0	1.1	2.3	3.1	5.6
**6**	4.5	6.0	2.5	2.6	1.1	0.0	2.5	4.6	9.8
**7**	7.3	13.4	4.8	4.9	2.3	2.5	0.0	5.3	17.4
**8**	8.2	15.5	5.7	6.2	3.1	4.6	5.3	0.0	10.8
**9**	10.9	23.5	9.1	10.2	5.5	9.8	17.4	10.8	0.0

**Table 3 ijerph-20-04960-t003:** Effort matrix $E_{2}$ for route two.

	1	2	3	4	5	6	7	8
**1**	0.0	0.9	2.2	2.4	5.5	9.0	9.9	13.9
**2**	0.9	0.0	2.1	2.5	7.5	16.6	18.6	26.9
**3**	2.2	2.1	0.0	1.2	4.4	7.9	9.3	14.3
**4**	2.4	2.5	1.2	0.0	1.1	2.5	3.3	5.7
**5**	5.5	7.5	4.4	1.1	0.0	2.5	4.7	9.9
**6**	9.0	16.6	7.9	2.5	2.5	0.0	5.5	17.4
**7**	9.9	18.6	9.3	3.3	4.7	5.5	0.0	10.7
**8**	13.9	26.9	14.3	5.7	9.9	17.4	10.7	0.0

**Table 4 ijerph-20-04960-t004:** Effort matrix $E_{3}$ for route three.

	1	2	3
**1**	0.0	10.2	15.7
**2**	10.2	0.0	6.8
**3**	15.7	6.8	0.0

**Table 5 ijerph-20-04960-t005:** Effort matrix $E_{4}$ for route four.

	1	2	3	4
**1**	0.0	5.2	5.6	19.6
**2**	5.2	0.0	8.4	20.1
**3**	5.6	8.4	0.0	11.7
**4**	19.6	20.1	11.7	0.0

**Table 6 ijerph-20-04960-t006:** Effort matrix $E_{5}$ for route five.

	1	2	3	4
**1**	0.0	13.4	15.3	18.4
**2**	13.4	0.0	4.6	8.8
**3**	15.3	4.6	0.0	4.6
**4**	18.4	8.8	4.6	0.0

**Table 7 ijerph-20-04960-t007:** Summary of routes.

	Route 1	Route 2	Route 3	Route 4	Route 5
**Ant’s efforts**	30.2	33.9	26.2	30.6	37.3
**Increased effort ** **(relative to the best route)**	1.153	1.294	1	1.168	1.424

## Data Availability

Not applicable.

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
