# Peer review of "Modified Ant Colony Optimization as a Means for Evaluating the Variants of the City Railway Underground Section"

_ijerph, 2023, doi:10.3390/ijerph20064960_

Round 1

Reviewer 1 Report

The paper proposes an interesting methodology for facing the transport network design. The method is based on the ant colony optimization and it is applied as support to design a city railway underground section.

Despite the potentialities of the paper, it presents some limits in relation to the state of the art and the methodology. In the follow, some broad and specific comments aimed at improving the paper’s quality.

Reviewer 2 Report

This paper presents a modern heuristic system for urban transport efficiency. It is a concrete application to underground rail routes in a real city. Thus, this modern algorithmization can be populated with real data. The results, discussion and conclusions are useful for both scientific and practical purposes. 

Reviewer 3 Report

This manuscript presented the modified classic ant colony optimization algorithm and its use in underground railway route design. Despite the modification sounds minor and average improvement, it still demonstrates interesting engineering merit. Overall, the paper was well written and clearly presented. It can be recommended for publication consideration, should the following minor comments can be considered and addressed.

1. The last sentence, quoted "The railway is one of the most energy-efficient modes of transport helping to enhance the environment and public health in the cities and agglomerations.", in the abstract section may need to be shifted to the begining? This more sounds like a background statement.

2. The past tense may need to be used in the abstract and other contexts of this manuscript? Now it mainly uses present and future tenses, which sounds a bit contradictory to common practices.

3. The title of the second section, quoted "2. Materials and Methods", sounds a bit weird, as this paper presented no tests but algorithms. Therefore, the use of "materials" seems irrelavant and may need to be changed.

4. Additional case studies may be provided if they are available to further substantiate this modified ant colony optimization algorithm; or althernatively, the comparison of this modidifed algorithm and other existing algorithms can be added to further highlight the originality, merit and value of this modification.

Round 2

Reviewer 1 Report

The revised version improves the paper's quality and readability.

Some final refinements are required for presenting the paper according to the standards of the journal

Author Response

Thank you for the opinion.

Our manuscript is written in accordance with the standards of the journal.